# Revealing the dynamic whole transcriptome landscape of *Clonorchis sinensis*: Insights into the regulatory roles of noncoding RNAs and microtubule-related genes in development

**Yangyuan Qiu[1☯], Cunzhou Wang[2☯], Jing Wang[1], Qingbo L. V.[1], Lulu Sun[1], Yaming Yang[3], Mingyuan Liu[1,4], Xiaolei Liu[1], Chen Li[1]\*, Bin Tang[1]\***

**1** State Key Laboratory for Diagnosis and Treatment of Severe Zoonotic Infectious Diseases, Key Laboratory for Zoonosis Research of the Ministry of Education, Institute of Zoonosis, and College of Veterinary Medicine, Jilin University, Changchun, PR China, **2** Jiashi County Hospital of Uygur Medicine, Kashgar City, Xinjiang Uygur Autonomous Region, PR China, **3** Yunnan Institute of Parasitic Diseases, Pu'er, PR China, **4** Jiangsu Co-innovation Center for Prevention and Control of Important Animal Infectious Diseases and Zoonoses, Yangzhou, PR China

☯ These authors contributed equally to this work.

\* lc2018@jlu.edu.cn (CL); tangbin1985@jlu.edu.cn (BT).

## Abstract

*Clonorchis sinensis* is a significant zoonotic food-borne parasite that causes a range of hepatobiliary diseases, which in severe cases can even lead to cholangiocarcinoma. To explore new diagnostic and treatment strategies, the dynamic RNA regulatory processes across different developmental stages of *C. sinensis* were analyzed by using whole-transcriptome sequencing. The chromosomal-level genome of *C. sinensis* was used for sequence alignment and annotation. In this study, we identified a total of 59,103 RNAs in the whole genome, including 2,384 miRNAs, 25,459 mRNAs, 27,564 lncRNAs and 3,696 circRNAs. Differential expression analysis identified 6,556 differentially expressed mRNAs, 2,231 lncRNAs, 877 miRNAs and 20 circRNAs at different developmental stages. Functional enrichment analysis highlighted the critical role of microtubule-related biological processes in the growth and development of *C. sinensis*. And coexpression analysis revealed 97 lncRNAs and 85 circRNAs that were coexpressed with 42 differentially expressed mRNAs that associated with microtubules at different developmental stages of *C. sinensis*. The expression of the microtubule-related genes dynein light chain 2 (DLC2) and dynein light chain 4 (DLC4) increased with *C. sinensis* development, and DLC2/4 could be inhibited by albendazole. Finally, by constructing competing endogenous RNA (ceRNA) networks, the lncRNA-miRNA-mRNA and circRNA-miRNA-mRNA regulatory relationships were constructed, and the ceRNA networks of MSTRG.14258.5-novel_miR_2287-newGene_28215 and MSTRG.14258.5-novel_miR_2216-CSKR_109340 were verified. This study suggests, through whole transcriptome sequencing, that the context of microtubule regulation may play an essential role in the development and growth of *C. sinensis*.

**Data Availability Statement:** The transcriptome data was submitted in the NCBI Sequence Read Archive and detail descriptive information can be viewed on NCBI's BioProject: PRJNA946132.

**Funding:** Pro. XL was supported by The National Key Research and Development Program of China (2021YFC2600200), Pro. ML was supported by Science and technology talents and platform plan of Yunnan province (Academician and Expert Workstation, 202305AF150167), and Drs. CL and BT were supported by the Fundamental Research Funds for the Central Universities. The funders had no role in study design, data collection and analysis, decision to publish, or preparation of the manuscript.

**Competing interests:** The authors have declared that no competing interests exist.

## Author summary

This study is the first to comprehensively elucidate the entire transcriptome of *Clonorchis sinensis* by exploring the expression and regulatory dynamics of noncoding RNAs and mRNAs at different developmental stages in its life cycle. By using next-generation sequencing technology, this research demonstrated new mRNAs, miRNAs, lncRNAs and circRNAs, and established valuable resources for future parasitic research. Compared with mRNAs, lncRNAs have shorter gene lengths, lower expression levels, fewer exons and fewer isoforms. Differential expression analysis demonstrated stage-specific patterns of noncoding RNAs and mRNAs in *C. sinensis* during its life cycle, thus providing crucial insights into the role of noncoding RNAs in parasite growth and development. Furthermore, the construction of ceRNA regulatory networks revealed that microtubule-related lncRNAs and circRNAs in *C. sinensis* may play an important role in its growth and development. This study advances our understanding of the entire transcriptome of *C. sinensis* and introduces novel perspectives for future parasitic infection treatment strategies. In summary, this study establishes a foundation for in-depth investigations into the interplay between noncoding RNAs and parasites, as well as the associated biological processes, thus demonstrating broad research and therapeutic potential.

## Introduction

*Clonorchis sinensis* is a food-borne parasite that is widely distributed in Southeast Asian countries, including China, South Korea and the Far East region of Russia [1]. *C. sinensis* parasitizes the liver and bile ducts of the host, thus resulting in a series of pathological changes, such as liver damage during the parasitic period; moreover, severe cases can develop into cirrhosis and cholangiocarcinoma [2–4]. It is estimated that approximately 13 million people in China are infected with *C. sinensis*, which has become a biological pathogen that threatens human survival and health [1].

C. sinensis, which is a parasitic organism, has a complex life cycle involving multiple hosts and stages. The developmental stages of this parasite include the egg, miracidium, sporocyst, redia, cercaria, metacercaria, juvenile and adult stages [5]. Previous studies have demonstrated distinct expression and regulatory patterns of genes across different developmental stages of parasites [6–8]. For instance, Yoo conducted an analysis of expressed sequence tags (ESTs) in three different developmental stages (adult, metacercaria and egg) of *C. sinensis* [9]. These findings indicate that each developmental stage is associated with distinct biological properties, growth characteristics, host adaptations and pathogenic features. Furthermore, the expression patterns of genes vary even among different tissues, thus highlighting the intricate regulatory mechanisms that can occur [10]. These results imply the crucial significance of transcriptome analysis in comprehending the survival strategies, respiratory processes, and metabolism of *C. sinensis* [11]. The differential gene expression and regulatory patterns across developmental stages and tissues provide valuable insights into the intricate biology of this parasite, thus providing information on potential targets for understanding its pathogenicity and for developing effective interventions.

Conventional transcriptomics primarily focuses on the expression levels of mRNAs; however, with technological advancements, the functional roles of noncoding RNAs (ncRNAs) are being increasingly recognized [12, 13]. Unlike mRNAs, ncRNAs constitute a class of RNA molecules that do not encode proteins, including microRNAs (miRNAs), long noncoding RNAs (lncRNAs) and circular RNAs (circRNAs). MiRNAs play a crucial role in gene

expression regulation by interacting with mRNAs, thereby modulating transcription and translation processes [14]. Additionally, lncRNAs exhibit complex and significant functions in cellular processes, such as the cell cycle, differentiation, and apoptosis. The diverse functions of these ncRNAs encompass various aspects, including cell signaling, gene expression balance, and disease development [15]. In studies related to human subjects, Maya utilized whole-blood transcriptome analysis in infants, and the results suggested the overexpression of interferon-stimulated genes and, to a lesser extent, of inflammation genes on day 7 [16]. In parasitic growth and developmental research, Mikhail compared two stages of *Opisthorchis felineus* parasitizing in different hosts and found significant differences in genes, G-protein coupled receptors and neuroactive signaling systems between the stages [6]. Although the miRNAs of *C. sinensis* have been sequenced, knowledge about lncRNAs and circRNAs remains elusive.

Given the intricate nature of biological networks, studies of individual mRNAs or ncRNAs lack correlations. Integrative analysis of multiple RNA information and exploration of potential network regulatory mechanisms have become trends in transcriptome research [13, 17]. Therefore, the whole transcriptome is an effective solution because it encompasses all transcripts that are transcribed from specific cells or tissues under certain temporal and spatial conditions, thus revealing essential biological regulatory patterns. Therefore, we utilized whole-transcriptome sequencing to profile ncRNAs at different stages of *C. sinensis*. This approach aimed to utilize comprehensive RNA-omics data to conduct differential expression analysis of ncRNAs, including the prediction of target genes, functional enrichment of target genes, and construction of networks of competing endogenous RNA (ceRNA).

## Materials and methods

### Ethics statement

All animals were handled strictly in accordance with the Animal Ethics Procedures and Guidelines of the People's Republic of China. The protocol was approved by the Institutional Animal Care and Use Committee of Jilin University (Protocol # 20200915).

### Sample collection and RNA extraction

*Pseudorasbora parva* were purchased from the market of *C. sinensis* endemic area in Fuyu city, Jilin Province. The meat of *P. parva* was mixed with gastric pepsin solution (containing 1% hydrochloric acid and 1% pepsin) at 37˚C for 3–4 h with continuous stirring with a magnetic stirrer. After digestion, clean metacercariae were selected under a stereomicroscope. Part of these metacercariae was used for excysted metacercariae samples, with samples collected one hour after postexcitation under 0.05% pancreatin; another portion was used for infecting rabbits, and adult worm and egg samples were collected. The samples of 5,000 metacercariae (M, n = 3), 5,000 excysted metacercariae (EM, n = 3), 30 adult worms (AD, n = 3) and 100,000 eggs (EGG, n = 3) were rapidly frozen in liquid nitrogen and stored at -80˚C until use. Total RNA was extracted by using the Qiagen RNeasy Mini Kit according to the manufacturer's instructions. For RNA quantification, RNA degradation and contamination (especially DNA contamination) were monitored on 1.5% agarose gels. RNA concentration and purity were measured by using a NanoDrop 2000 Spectrophotometer (Thermo Fisher Scientific, Wilmington, DE, USA). RNA integrity was assessed by using the RNA Nano 6000 Assay Kit of the Agilent Bioanalyzer 2100 System (Agilent Technologies, CA, USA).

## Library construction and sequencing

To construct the library of mRNA, lncRNA and circRNA of *C. sinensis*, ribosomal RNAs (rRNAs) were removed by using the Ribo-Zero rRNA Removal Kit (Epicenter, Madison, WI, USA). After following the manufacturer's guidelines, sequencing libraries were created with the NEBNext Ultra Directional RNA Library Prep Kit for Illumina (NEB, USA), and index codes were incorporated to attribute sequences to individual samples. To isolate insert fragments within the 150 to 200 bp range, the library fragments were purified by using AMPure XP Beads (Beckman Coulter, Beverly, MA, USA). A 3 μl volume of USER Enzyme (NEB, USA) was applied to size-selected, adaptor-ligated cDNA at 37°C for 15 min before PCR, which was performed by using Phusion High-Fidelity DNA polymerase. Finally, the PCR products were purified by AMPure XP system, and the library quality was evaluated on an Agilent Bioanalyzer 2100 and by RT–qPCR. For the *C. sinensis* small RNA library, 2.5 ng of RNA per sample was utilized for RNA sample preparation. Sequencing libraries were generated by using the NEBNext Ultra Small RNA Sample Library Prep Kit for Illumina (NEB, USA) according to the manufacturer's recommendations, and index codes were added for sequence mapping. PAGE was employed for fragment screening, and small RNA libraries were obtained through rubber cutting and recycling. Finally, the PCR products were purified by AMPure XP system and the library quality was assessed on an Agilent Bioanalyzer 2100 system.

Clustering of the index-coded samples was performed on a cBot Cluster Generation System by using the TruSeq PE Cluster Kit v3-cBot-HS (Illumina) according to the manufacturer's instructions. After cluster generation, the library preparations were sequenced on an Illumina platform, and reads were generated.

## Identification of *C. sinensis* ncRNAs

To identify lncRNAs of *C. sinensis*, the transcriptome was assembled by using StringTie based on the reads that were mapped to the reference genome (GenBank: GCA_003604175.2; isolate: Cs-k2). The assembled transcripts were annotated by using the GffCompare program. The unknown transcripts were used to screen for putative lncRNAs. Four computational approaches, including CPC/CNCI/Pfam/CPAT, were combined to sort non-protein-coding RNA candidates from putative protein-coding RNAs in unknown transcripts. For the identification of *C. sinensis* circRNAs, CIRI (CircRNA Identifier) tools, which scanned SAM files twice and collected sufficient information to identify and characterize circRNAs, were used to identify circRNAs. For miRNAs, the clean reads were aligned with the GtRNAdb, Rfam and Repbase databases, as well as filtered rRNA, transfer RNA (tRNA), small nuclear RNA (snRNA), small nucleolar RNA (snoRNA), and other repeats. The remaining reads were used to recognize known miRNAs and novel miRNAs predicted by comparison with known miRNAs from miRBase.

## Quantitative real-time PCR

To verify the stability of the sequencing results from two independent libraries, we randomly selected three sequencing outcomes from both the lncRNA library and the miRNA library and performed quantitative real-time PCR (RT–qPCR) to verify their expression levels. Reverse transcription of miRNA was performed by using a miRNA 1st Strand cDNA Synthesis Kit (Vazyme, China) and a PrimeScript RT Reagent Kit with gDNA Eraser (TaKaRa, Japan). RT–qPCR was performed by using 2× RealStar Fast SYBR qPCR Mix (High ROX) (Kangrun Biology Co., Ltd., China), and the reaction conditions were as follows: initial incubation at 95°C for 3 min, followed by 40 cycles of 15 s (denaturation) at 95°C and 30 s (annealing and extension). In addition, 18S ribosomal RNA was used as an internal reference for the lncRNA

sequence, and the U6 gene was used for the miRNA sequence. The primer sequences are presented in S1 Table.

## Differential expression analyses

Gene expression levels were quantified through the calculation of fragments per kilobase of transcript per million fragments that were mapped (denoted as FPKM). Differential expression analysis between two groups was conducted by using the DESeq2 R package (version 3.2.0). The resulting $P$ values were adjusted by using the Benjamini and Hochberg approach to control the false discovery rate. Genes exhibiting an adjusted $P$ value $< 0.001$ and an absolute $\log_2$(fold change) $> 1$, as determined via DESeq2, were classified as being differentially expressed.

## Enrichment analysis of Gene Ontology Biological Process (GO-BP) and Kyoto Encyclopedia of Genes and Genomes (KEGG)

GO-BP analysis of the differentially expressed genes (DEGs) was conducted by using the GOseq R package (version 1.24.0), which employs the Wallenius noncentral hypergeometric distribution for enrichment analysis [18]. This method is particularly advantageous because it adjusts for gene length bias in DEGs. For a comprehensive understanding of high-level functions and utilities within the biological system, especially in *C. sinensis*, we leveraged the KEGG database resource [19]. KEGG is invaluable for interpreting large-scale molecular datasets that are generated through genome sequencing and other high-throughput experimental technologies (http://www.genome.jp/kegg/). To assess the statistical enrichment of DEGs in KEGG pathways, we utilized KOBAS software (version 3.0) [20]. These analytical approaches collectively contribute to a robust exploration of the functional significance of DEGs in the context of biological pathways.

## Verification of the relationship between microtubule-related genes and growth

To investigate growth-related genes, we validated ten mRNA genes that have biological functions associated with microtubules. Rabbits were infected with 500 metacercariae per rabbit, and parasites were collected at day 7, 14 and 56 post-infection. On the 7th and 14th days after infection, the rabbits were euthanized, and the liver tissues were separated and washed three times with PBS. First, the liver was cut along the bile duct and slowly compressed toward the common bile duct from the edge of the liver. Second, the liver was cut into small pieces and placed into a dish containing PBS. The larger tissue blocks were removed, and the parasites were collected under a stereomicroscope. The rabbits were euthanized on the 56th day after infection. Afterwards, the liver was separated and washed three times with PBS, and the liver was slowly compressed toward the common bile duct from the edge of the liver. Subsequently, the worms were squeezed out of the common bile duct and collected. RNA extraction was performed according to the kit instructions, and reverse transcription was performed by using an RNA reverse transcription kit. RT–qPCR was performed to measure the expression of microtubule-related genes.

In addition, as albendazole affects the cytoplasmic microtubule system of the parasite's intestinal wall cells, we explored changes in the expression of microtubule-associated genes after albendazole treatment. Adult *C. sinensis* collected at the 30th day post-infection were equilibrated in high sugar DMEM, washed and incubated with or without albendazole. After

24 hours, RNA of *C. sinensis* was extracted and reverse transcribed, and RT–qPCR was performed on the collected worms treated with or without albendazole.

### miRNA response element (MRE) identification

To investigate the mRNA/lncRNA/circRNA binding sites for MREs, we used RNAhybrid [21], miRanda [22], and PITA [23] to identify the binding pairs between miRNAs and mRNAs/lncRNAs/circRNAs. As previously described in *Fasciola hepatica* [24], all of the mRNA-miRNA, lncRNA-miRNA and circRNA-miRNA pairs must meet the three algorithm thresholds of RNAhybrid, $P < 0.1$, energy $< -22$; miRanda, total score $>145$, energy $< -10$; and PITA, $\Delta\Delta G < -10$.

### Construction and verification of the ceRNA network

By utilizing the earlier predictions of miRNA-mRNA, lncRNA-miRNA and circRNA-miRNA interaction pairs, we assembled lncRNA-miRNA-mRNA or circRNA-miRNA-mRNA triads that exhibited shared miRNA components. At the same time, we identified mRNAs and lncRNA/circRNA interaction pairs with a high coexpression correlation ($|r| > 0.9$) for subsequent inclusion in the construction of the ceRNA network. Finally, the ceRNA network was drawn with Cytoscape v3.10.1 (https://cytoscape.org/).

To verify the role of the ceRNA network in the growth and development of *C. sinensis*, we validated the lncRNA MSTRG.14258.5 in the ceRNA network. First, to interfere with the expression of MSTRG.14258.5, we designed three small interfering RNAs (siRNAs) via DSIR (http://biodev.extra.cea.fr/DSIR/DSIR.html), and the sequences of the utilized siRNAs are listed in S2 Table. Afterwards, *C. sinensis* of 56 day were used for electroporation. Twenty adult *C. sinensis* were dispensed into an electroporation cuvette with a 4 mm gap (Bio-Rad, Hercules, CA) containing 5 µM siRNA in a total volume of 500 µl of DMEM and subjected to exponential decay pulses at 200 V and 800 µF for 20 ms (Gene Pulser Xcell, Bio-Rad). The electroporated *C. sinensis* were maintained in culture in DMEM supplemented with siRNA for 48 hours. After 48 hours, RNA was extracted from the parasites, and RT–qPCR was used to detect the expression of miRNAs and mRNAs in the ceRNA network of MSTRG.14258.5.

## Results

### Genome chromosome alignment and characteristics of the whole transcriptome

To annotate the sequences that were obtained from sequencing, we aligned the reads with the chromatin-level genome (CSKR.v2, GCA_003604175.2) to determine the genomic positions of various sequence types, as presented in Fig 1A to 1D. Only seven chromosomes (CM030369.1, CM030370.1, CM0303671.1, CM030372.1, CM030373.1, CM030374.1 and CM030375.1) were aligned with mRNAs, lncRNAs, and miRNAs. In addition to the seven chromosomes aligned with mRNAs, lncRNAs and miRNAs, some scaffolds, such as NIRI02000004.1, NIRI02000025.1 and NIRI02000028.1, could also be aligned with circRNAs. Notably, Fig 1B displays the chromosomal positions of four lncRNA types (lincRNAs, antisense lncRNAs, intronic lncRNAs and sense lncRNAs), and the overlapping positions suggested potential sequence overlap among the four lncRNA types.

Through database comparison and software prediction, we identified and analyzed the diverse RNA types in the entire genome of *C. sinensis*. In total, 59,103 RNAs were identified, comprising 2,384 miRNAs, 25,459 mRNAs, 27,564 lncRNAs, and 3,696 circRNAs (Fig 1E). The length distribution of the miRNAs ranged from 18 bp to 25 bp, with a predominant length

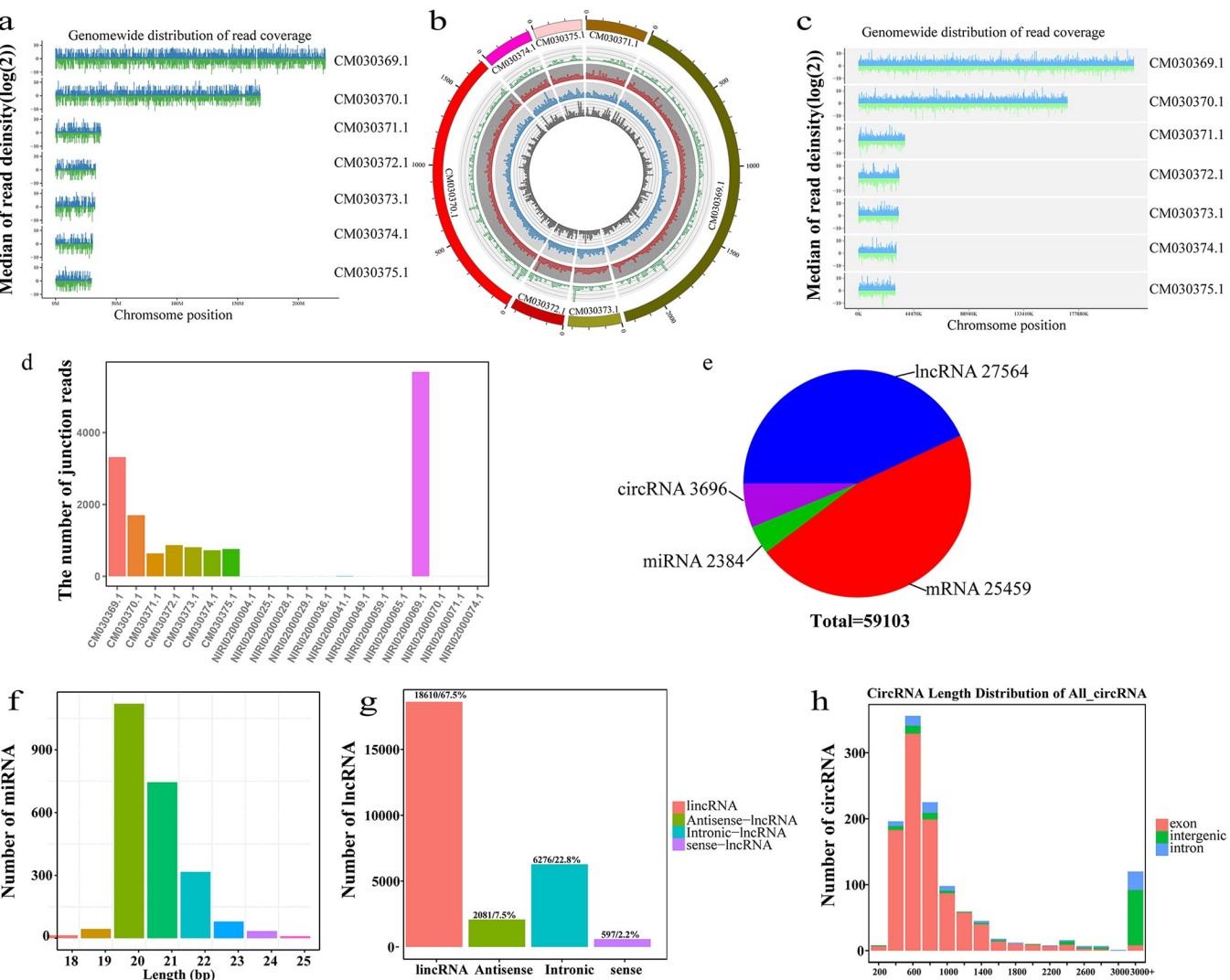

**Fig 1. Alignments of whole-transcriptome RNAs and chromosomes and the characteristics of noncoding RNAs in *C. sinensis*.** (**a**) The distribution of miRNA reads along the reference genome chromosome was analyzed in terms of position and coverage depth. (**b**) Different categories of lncRNAs exhibit different distribution patterns on the reference genome chromosome. The outermost layer represents the chromosome ring of genome, including sense lncRNA (green), intergenic lncRNA (lincRNA) (red), antisense lncRNA (gray) and intron lncRNA (blue). (**c**) The position and coverage depth distribution of mRNA reads on the reference genome chromosome. (**d**) The number of circRNA RNA on the reference genome chromosomes. NIRI02 is the WGS project for genome sequencing of *C. sinensis*. (**e**) The counts of distinct RNA types in *C. sinensis*. (**f**) The number of different length miRNAs. (**g**) The abundance and proportion of different lncRNA types. (**h**) The number of different type circRNAs.

of 20–22 bp (Fig 1F), which is consistent with typical miRNA characteristics. Among the four types of lncRNAs, lincRNAs were the most abundant (18,610, 67.5%), whereas sense lncRNAs were the least abundant (597, 2.2%) (Fig 1G). The lengths of circRNAs were predominantly between 400 bp and 1,400 bp, with exon circRNAs representing the greatest proportion (Fig 1H). These results indicated that different types of noncoding RNAs can be effectively annotated to the genome of *C. sinensis*.

## Comparison of lncRNAs and mRNAs

According to Fig 1E, there were 27,564 lncRNAs and 25,459 mRNAs in *C. sinensis*. By comparing the characteristics of lncRNAs and mRNAs in *C. sinensis*, we found that they differed in

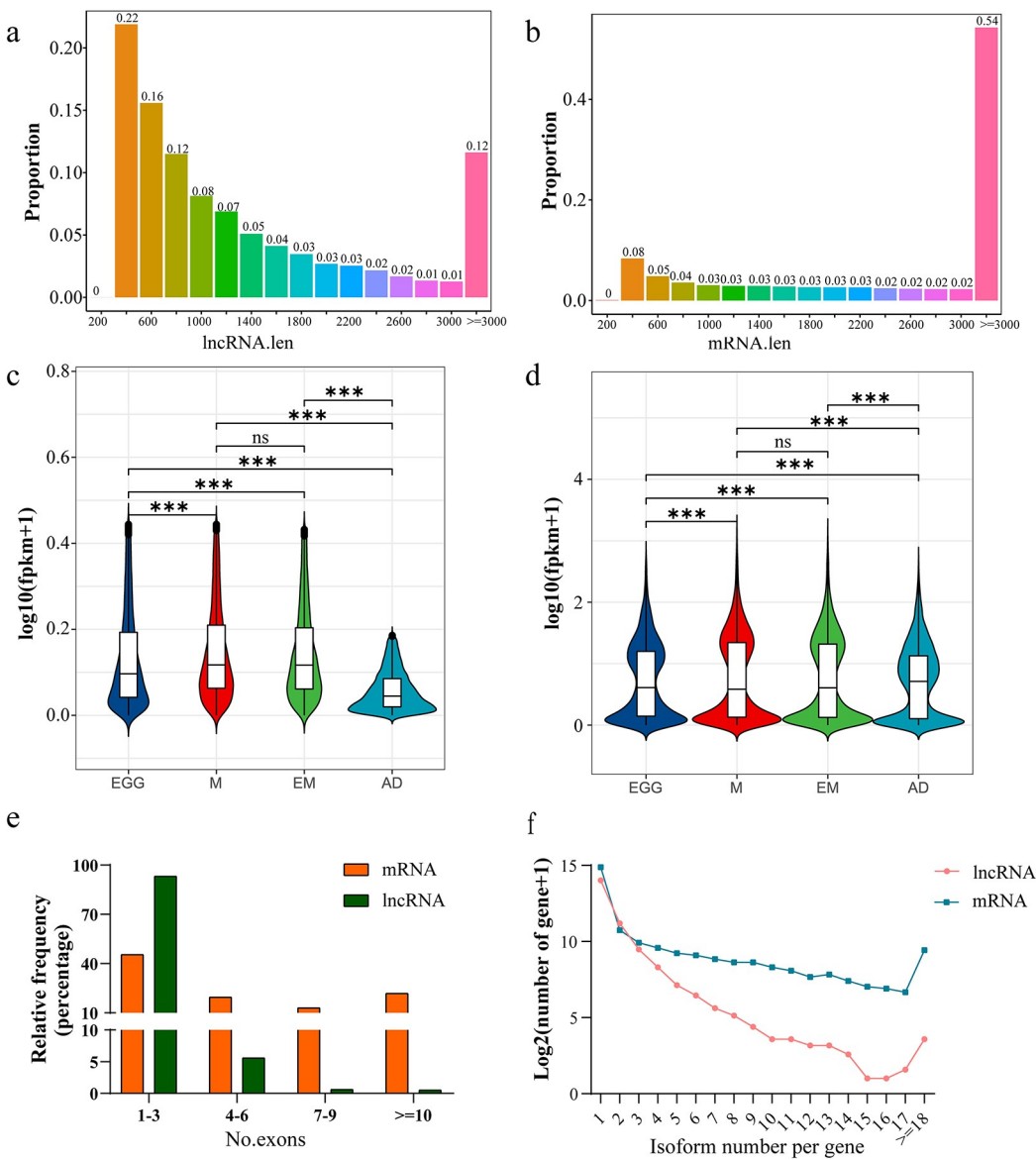

**Fig 2. Comparison of mRNAs and lncRNAs in *C. sinensis*.** (**a**) Proportion of lncRNAs of different lengths. (**b**) Proportion of mRNAs of different lengths. (**c**) LncRNA expression levels in *C. sinensis* at different developmental stages. EGG is egg, M is metacercariae, EM is excysted metacercariae and AD is adult worm. (**d**) mRNA expression levels in *C. sinensis* at different developmental stages. (**e**) Percentage distribution of distinct exons in lncRNAs and mRNAs. (**f**) Number of isoforms per gene. Compared with indicated groups, ns is for no significance, *** is for *P* < 0.001.

length, gene expression, number of exons and isoform number per gene (Fig 2). The lncRNA transcripts were generally shorter and mostly localized between 200–1,000 bp (Fig 2A), whereas mRNAs included the majority of genes (54%), with a length greater than or equal to 3,000 bp (Fig 2B). In addition, compared to those of mRNAs, the expression levels of lncRNAs were very low. At different developmental stages, mRNAs and lncRNAs were differentially expressed; however, there was no significant difference between metacercariae and excysted metacercariae (Fig 2C and 2D). Compared to the number of mRNA exons (median = 4 exons, Fig 2E), lncRNA genes contained fewer exons (median = 2 exons; Fig 2E). Similarly, lncRNAs had fewer isoforms per gene (Fig 2F). These results suggested that compared to mRNAs,

lncRNAs of *C. sinensis* have shorter gene lengths, lower expression levels, fewer exons and fewer isoforms.

## Differential expression analysis

For the identification of differentially expressed RNAs, we applied screening criteria ($|\log_2(\text{Fold change})| > 1$ and adjusted $P < 0.001$) to identify differentially expressed RNAs. A total of 6,556 differentially expressed mRNAs (DE-mRNAs) were identified, comprising 3,717 upregulated and 2,839 downregulated mRNAs at four stage comparisons (Fig 3A and S3 Table). A total of 2,231 differentially expressed lncRNAs (DE-lncRNAs) were identified, with 1,600 upregulated and 631 downregulated lncRNAs at four stage comparisons (Fig 3B and S4 Table). Furthermore, 877 differentially expressed miRNAs (DE-miRNAs) were identified, including 477 upregulated and 400 downregulated miRNAs at four stage comparisons (Fig 3C and S5 Table). Additionally, a total of 20 differentially expressed circRNAs (DE-circRNAs) were found, with 11 upregulated and 9 downregulated circRNAs at four stage comparisons (Fig 3D and S6 Table). To validate the accuracy of the sequencing outcomes, a random selection of miRNAs and lncRNAs was subjected to RT–qPCR validation. The RT–qPCR results showed

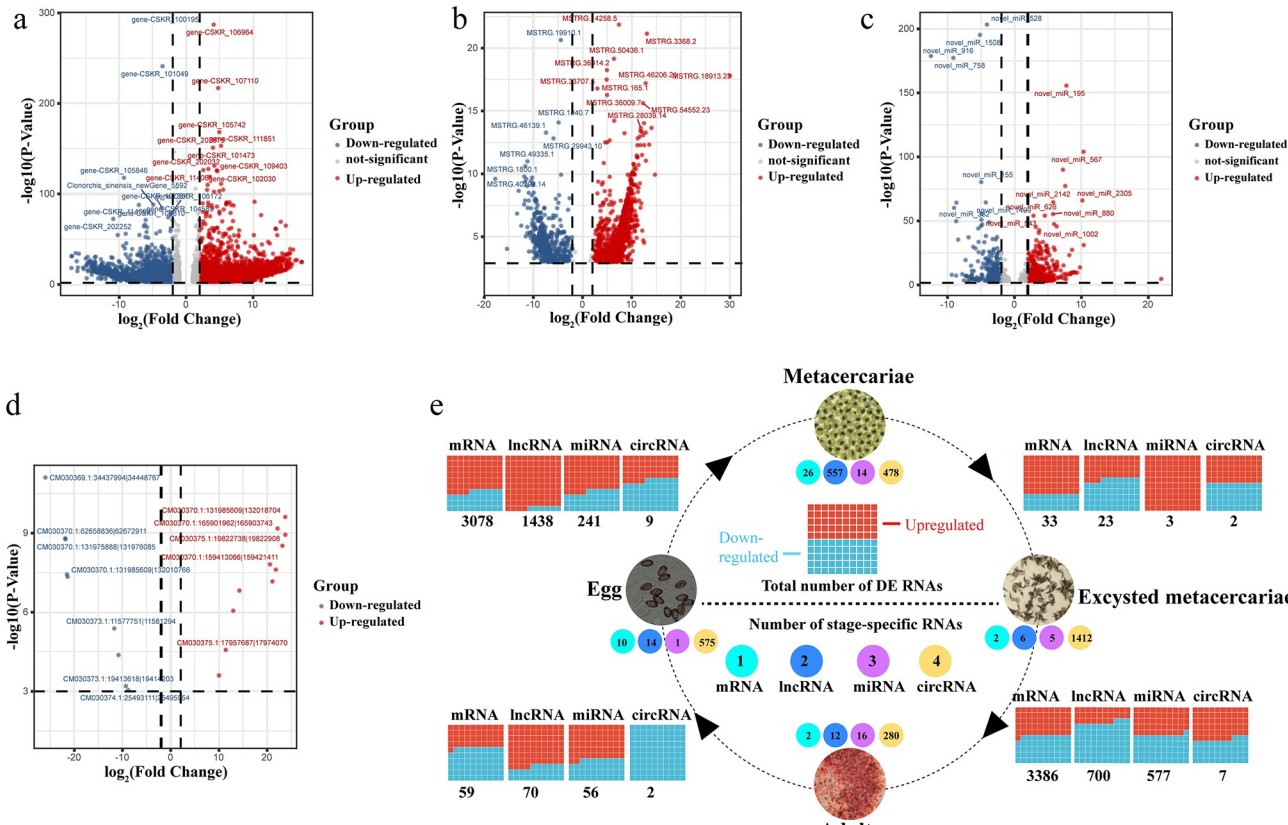

**Fig 3. Differential expression (DE) of the developmental regulation of noncoding RNAs and mRNAs in the four stages of *C. sinensis* development.** (**a**) Fold change and adjusted *P* value cutoff (*P* < 0.001) of DE mRNAs at different stages. (**b**) Fold change and adjusted *P* value cutoff (*P* < 0.001) of DE lncRNAs at different stages. (**c**) Fold change and adjusted *P* value cutoff (*P* < 0.001) of DE miRNAs at different stages. (**d**) Fold change and adjusted *P* value cutoff (*P* < 0.001) of DE circRNAs at different stages. (**e**) Summary data from the DE analysis of transcripts in metacercariae, excysted metacercariae, adults and eggs. The numbers of stage-specific ncRNAs and mRNAs are specified for each life stage, along with the counts of DE ncRNA and mRNA transcripts, and the proportions of upregulated and downregulated transcripts during each developmental transition are also presented. The comparisons at different stages are egg::metacercariae, metacercariae::excysted metacercariae, excysted metacercariae::adult and adult::egg.

similar trend to the whole-transcriptome sequencing results (S1 Fig), which confirmed the reliability of the sequencing data.

For a comprehensive representation of gene expression across development stages of *C. sinensis*, Fig 3E depicts the number and proportion of DEGs at different developmental stages. Notably, the number of DEGs from the excysted metacercaria stage to the adult stage and from the egg stage to the metacercaria stage was greater than that from the other stages. Furthermore, stage-specific expression analysis demonstrated unique RNAs for different developmental stages of *C. sinensis*. Specifically, in the adult stage, there were two specifically expressed mRNAs, 12 lncRNAs, 16 miRNAs and 280 circRNAs (Fig 3E). Remarkably, circRNAs exhibited the strongest stage-specific expression, with 478, 1,412, 280, and 575 expression numbers in metacercariae, excysted metacercariae, adults and eggs, respectively (Fig 3E). The expression of DEGs at different stages indicated that the RNAs of *C. sinensis* remained in a dynamic state with growth and development.

## Functional enrichment analysis of DE-mRNAs

To determine the mRNAs associated with growth and development, we performed GO-BP functional enrichment analysis for DE-mRNAs. The results demonstrated that numerous DE-mRNAs were significantly enriched in microtubule-related biological processes, including microtubule-based processes, movements based on microtubules and bundle formation (Fig 4A–4C). Interestingly, for the DE-mRNAs in the metacercariae and excysted metacercariae stages, the GO-BP terms were primarily related to three biological processes: response to

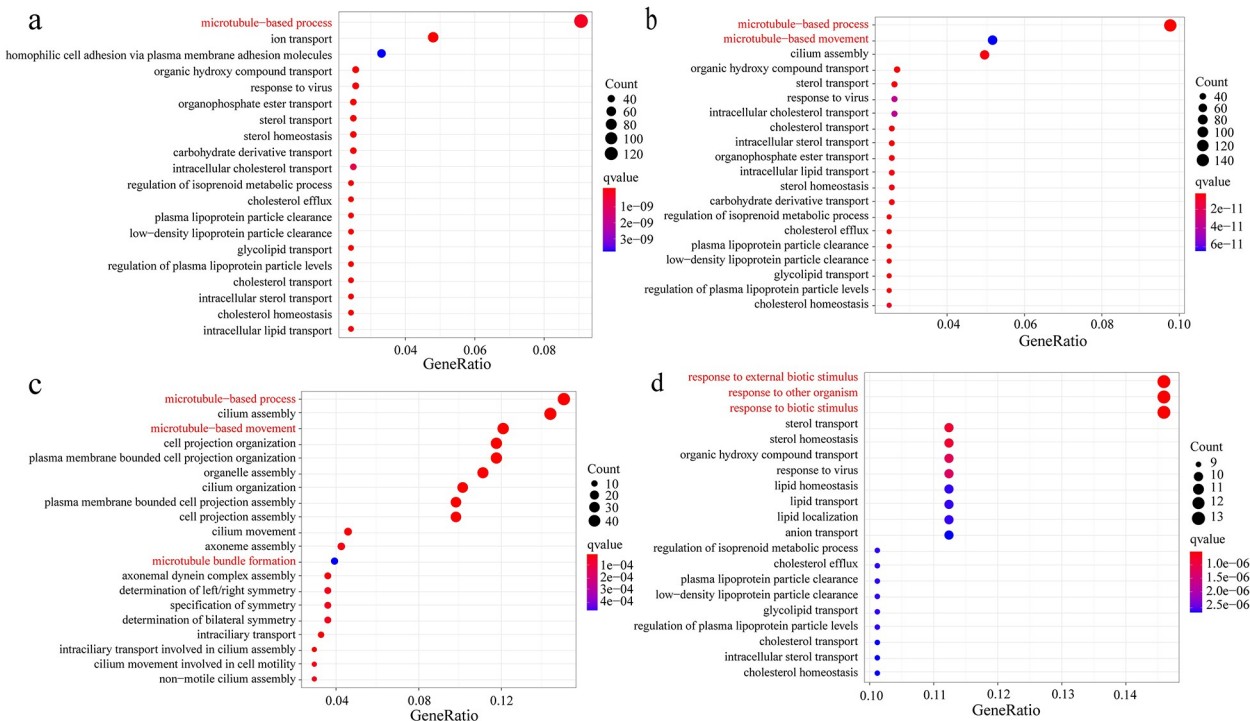

**Fig 4. Enrichment of Gene Ontology Biological Process (GO-BP) of differentially expressed mRNAs in four developmental stages of *C. sinensis*.** The top 20 pathways or GO-BP terms enriched in the differentially expressed mRNAs. (**a**) The top 20 GO-BP terms enriched between adults and metacercariae in *C. sinensis*. (**b**) The top 20 GO-BP terms enriched between adult and excysted metacercariae in *C. sinensis*. (**c**) The top 20 GO-BP terms enriched between adults and eggs in *C. sinensis*. (**d**) The top 20 GO-BP terms enriched in metacercariae and excysted metacercariae in *C. sinensis*.

external biological stimulus, response to other organisms and response to biological stimulus (Fig 4D). Given that the observed enrichment highlights the importance of microtubule-related biological processes in the growth and development of *C. sinensis*, subsequent analyses should focus on exploring the regulatory role of microtubule-related RNAs in *C. sinensis* growth and development.

## Annotation and validation of microtubule-related genes

To identify differential mRNAs associated with microtubules in the three stages (AD::M, AD::EM and AD::EGG) of *C. sinensis*, we compared DE-mRNAs associated with microtubule-related GO-BP terms. Forty-two DE-mRNAs were consistently present across all three stages (Fig 5A). Coexpression analysis revealed 97 lncRNAs and 85 circRNAs that were coexpressed with these 42 DE-mRNAs (Fig 5A and S7 Table), which suggested potential regulatory roles for these lncRNAs and circRNAs in *C. sinensis* microtubule regulation. Functional analysis of proteins encoded by the 42 DE-mRNAs through the KEGG analysis revealed that 20 of them were associated with the cytoskeleton, predominantly including tubulin alpha chain, dynein light chain, and hypothetical protein genes (Fig 5A). Several other hypothetical proteins and functionally uncharacterized protein genes have also been implicated in microtubule regulation.

To validate the differential expression of these genes during the growth and development of *C. sinensis*, ten genes were selected for RT–qPCR analysis. The results demonstrated that dynein light chain 2 (DLC2) and dynein light chain 4 (DLC4) exhibited increasing expression as the parasite developed (Fig 5B). Additionally, treatment of adult *C. sinensis* with albendazole demonstrated that at a concentration of 1 μg/ml, DLC2 did not differ between the drug-treated and nondrug-treated groups, whereas DLC4 expression significantly decreased (Fig 5C). At a concentration of 10 μg/ml albendazole, both DLC2 and DLC4 exhibited significant decreases in gene expression in the drug-treated group (Fig 5C). Thus, DLC2 and DLC4 may be important for the growth and development of *C. sinensis* and may be used as drug targets.

## Construction of the ceRNA network

By utilizing the regulatory relationships between DE-miRNAs and DE-mRNAs/DE-lncRNAs, we constructed ceRNA regulatory networks at different developmental stages, as shown in S8 Table. In EM::AD comparasion, 53 lncRNA-miRNA-mRNA interactions were identified, whereas M::AD comparasion exhibited 296 interactions, and AD::EGG comparasion demonstrated ten interactions. Subsequently, by leveraging lncRNA and microtubule related mRNA coexpression relationships, lncRNAs MSTRG.14258.5, MSTRG.19374.4 and MSTRG.9295.1 were speculated to be possibly related to microtubule regulation. Furthermore, the resulting ceRNA regulatory networks involving these three lncRNAs were plotted (Fig 6). Within the lncRNA-regulated ceRNA network, the three lncRNAs collectively regulated 68 miRNAs, and these miRNAs correspondingly regulated 11 mRNAs (S9 Table).

Similarly, by exploiting the regulatory relationships between DE-miRNAs and DE-mRNAs/circRNAs, we constructed ceRNA regulatory networks at different developmental stages (S10 Table). In EM::AD comparasion, ten circRNA-miRNA-mRNA interactions were identified, 24 circRNA-miRNA-mRNA interactions were identified in M::AD comparasion and three circRNA-miRNA-mRNA interactions were identified in EGG::AD comparasion. Regarding the coexpression relationship of circRNAs and microtube-related mRNA genes, CM030371.1:20203323|20355085 and CM030369.1:93269800|93305609 were identified as being circRNAs related to microtubule regulation. The resulting ceRNA regulatory network involving two circRNAs was mapped (Fig 7). In the circRNA-regulated ceRNA network, the

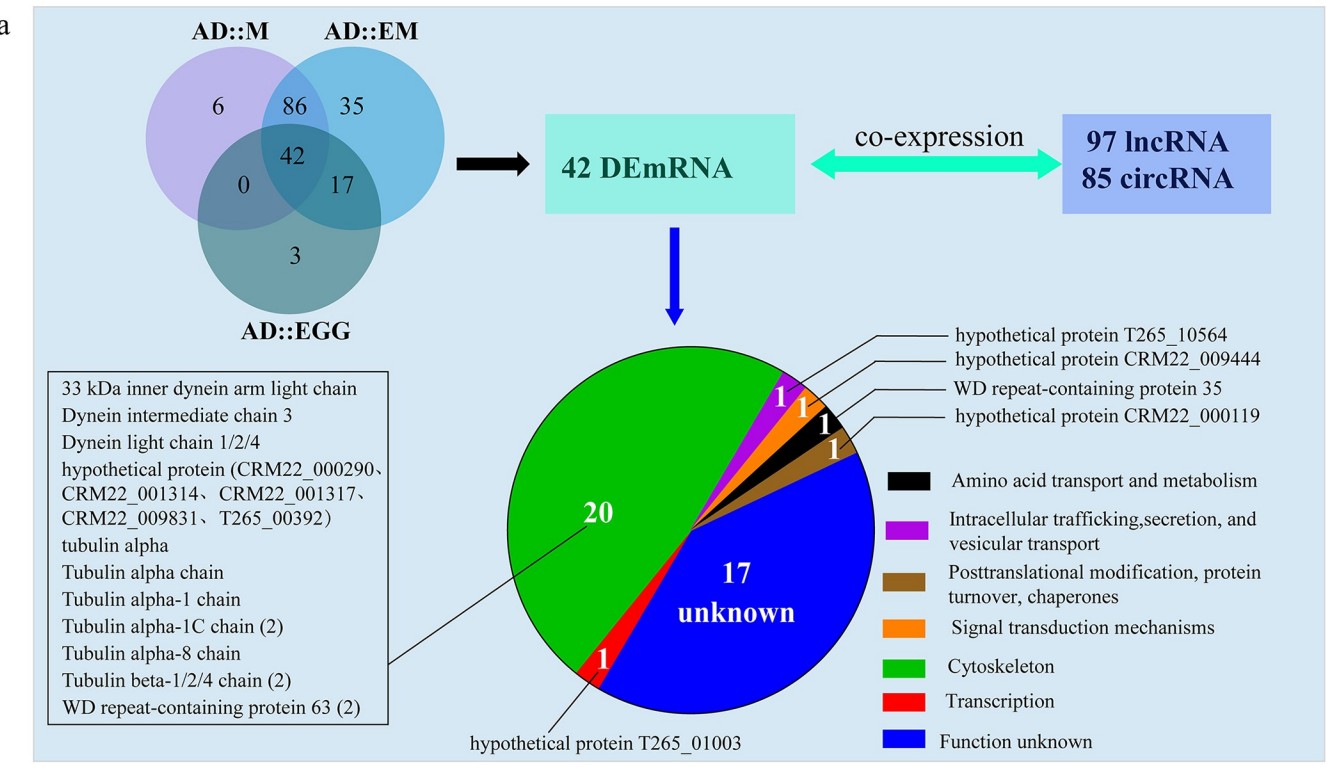

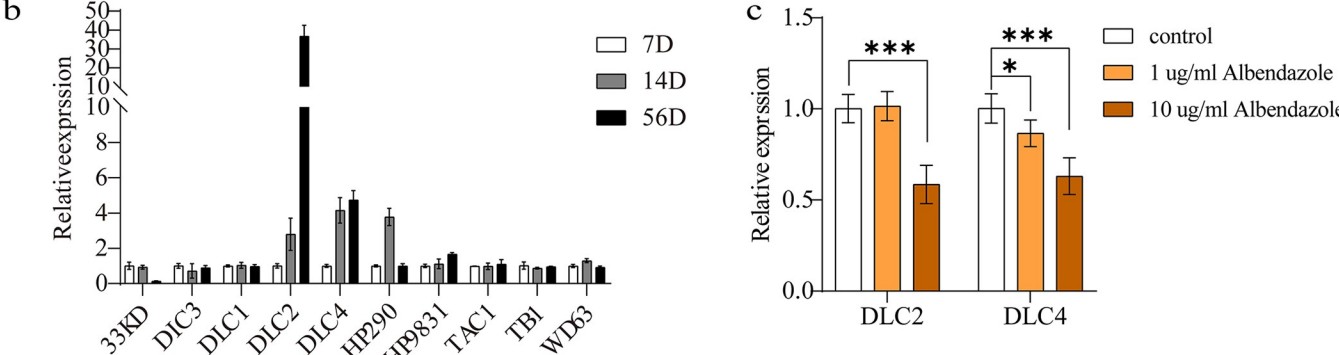

**Fig 5. Coexpression of lncRNAs/circRNAs and 42 microtubule-related mRNAs in *C. sinensis* and verification of microtubule-related mRNAs.** (**a**) Global coexpression analysis of all lncRNAs/circRNAs and mRNAs in our dataset identified 97 lncRNAs and 85 circRNAs that were coexpressed with 42 microtubule-related mRNAs. (**b**) The expression of ten cytoskeleton genes at different developmental stages of *C. sinensis* (the 7th day, 14th day and 56th day) using the 18S ribosomal RNA gene as an internal reference. 33KD is 33 kDa inner dynein arm light chain; DIC3 is dynein intermediate chain 3; DLC2 is dynein light chain 2; DLC4 is dynein light chain 4; HP290 is hypothetical protein CRM22_000290; HP9831 is hypothetical protein CRM22_009831; TAC1 is tubulin alpha-1c; TB1 is tubulin beta-1; WD63 is WD repeat-containing protein 63. (**c**) The expression of DLC2 and DLC4 after *C. sinensis* adults were treated with albendazole. Compared with indicated groups, * is for $P < 0.05$, *** is for $P < 0.001$.

two circRNAs collectively regulated 73 miRNAs, and these miRNAs correspondingly regulated 10 mRNAs (S11 Table). Crucially, miRNAs that act as hubs in the lncRNA/circRNA-miRNA-mRNA ceRNA networks are highlighted, with the more critical miRNAs (top five degrees) marked in red in Figs 6 and 7. The regulatory networks of lncRNAs and circRNAs indicated that different types of noncoding RNAs in *C. sinensis* can be linked through the ceRNA network, thereby regulating the growth and development of *C. sinensis*.

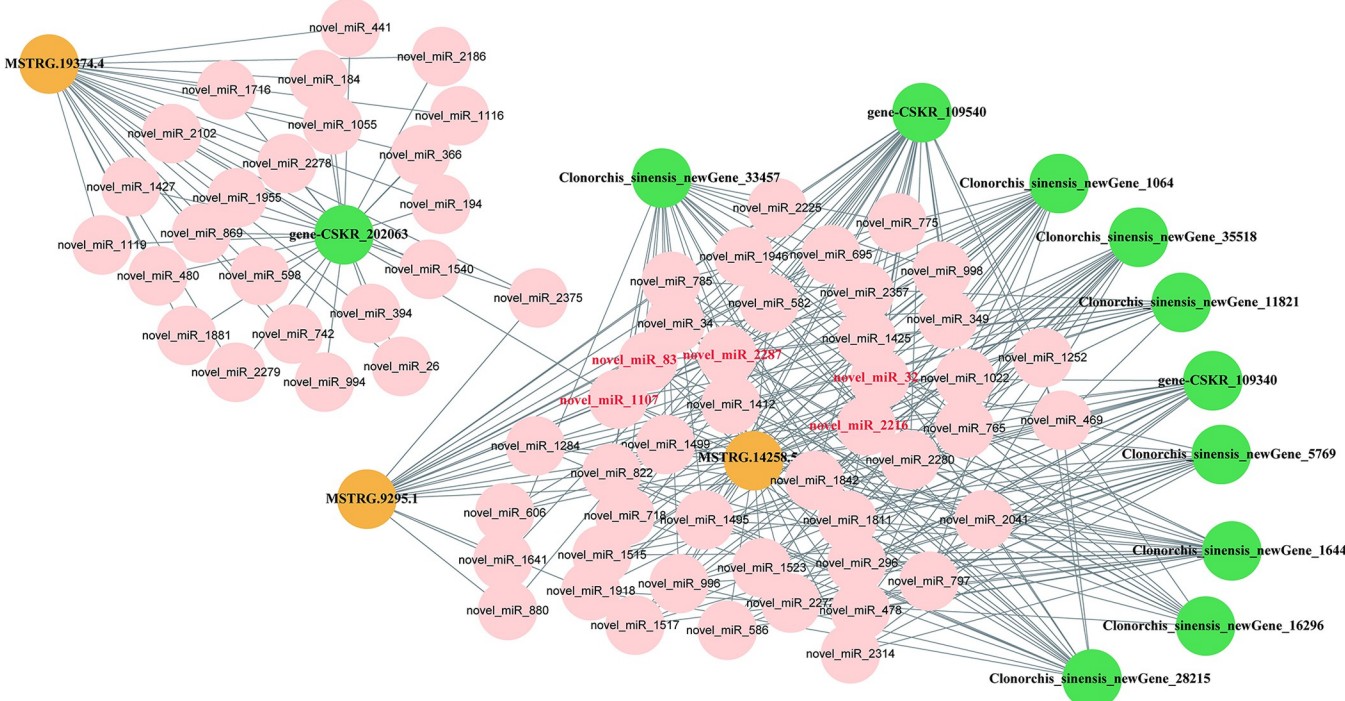

**Fig 6. ceRNA network representation of computationally predicted binding interactions between lncRNAs, miRNAs and mRNAs in *C. Sinensis*.** This network represents an excerpt of the lncRNA-miRNA-mRNA network detailed in S9 Table, showing only the lncRNAs that coexpressed microtubule-related mRNAs in *C. sinensis*. The top5 degree miRNAs are novel_miR_83, novel_miR_2287, novel_miR_32, novel_miR_2216, and novel_miR_1107 in lncRNA ceRNA. The lncRNA nodes are represented by yellow circles, the miRNA nodes are represented by tangerine circles, and the mRNA nodes are represented by green circles. Connections between nodes are represented by gray edges.

## Verification of the lncRNA ceRNA network

To demonstrate that noncoding RNAs can participate in gene expression regulation during the growth process of *C. sinensis* through a ceRNA mechanism, we selected the lncRNA MSTRG.14258.5 for validation. Fig 8A shows that MSTRG.14258.5 can regulate five miRNAs, which correspondingly regulate ten mRNAs. In addition, by interfering with MSTRG.14258.5, the expression of target miRNAs and mRNAs was evaluated. Fig 8B shows that the FAM-labeled siRNA can be electrotransferred to the cortex of *C. sinensis*. In addition, we also detected a large amount of the labeled siRNA in the intestine of *C. sinensis*, which was likely due to *C. sinensis* ingests it via feeding. By comparing the interference effects of the three siR-NAs, we found that siRNA#2 had the greatest interference effect (Fig 8C). Therefore, we chose siRNA#2 for subsequent interference experiments. After 48 hours of MSTRG.14258.5 interference, we detected changes in the expression of the target genes of the miRNAs, as shown in Fig 8D, which indicated that novel_miR_2287 and novel_miR_2216 could be regulated by MSTRG.14258.5. In addition, the changes in mRNAs regulated by novel_miR_2287 and novel_miR_2216 are shown in S2 Fig. We found that lncRNA (MSTRG.14258.5)-miRNA (novel_miR_2287)-mRNA (newGene_28215) and lncRNA (MSTRG.14258.5)-miRNA (novel_miR_2216)-mRNA (CSKR_109340) significantly changed after lncRNA MSTRG interference (shown in Fig 8E and 8F), which suggested that the ceRNA networks are real in the regulation of growth and development in *C. sinensis*.

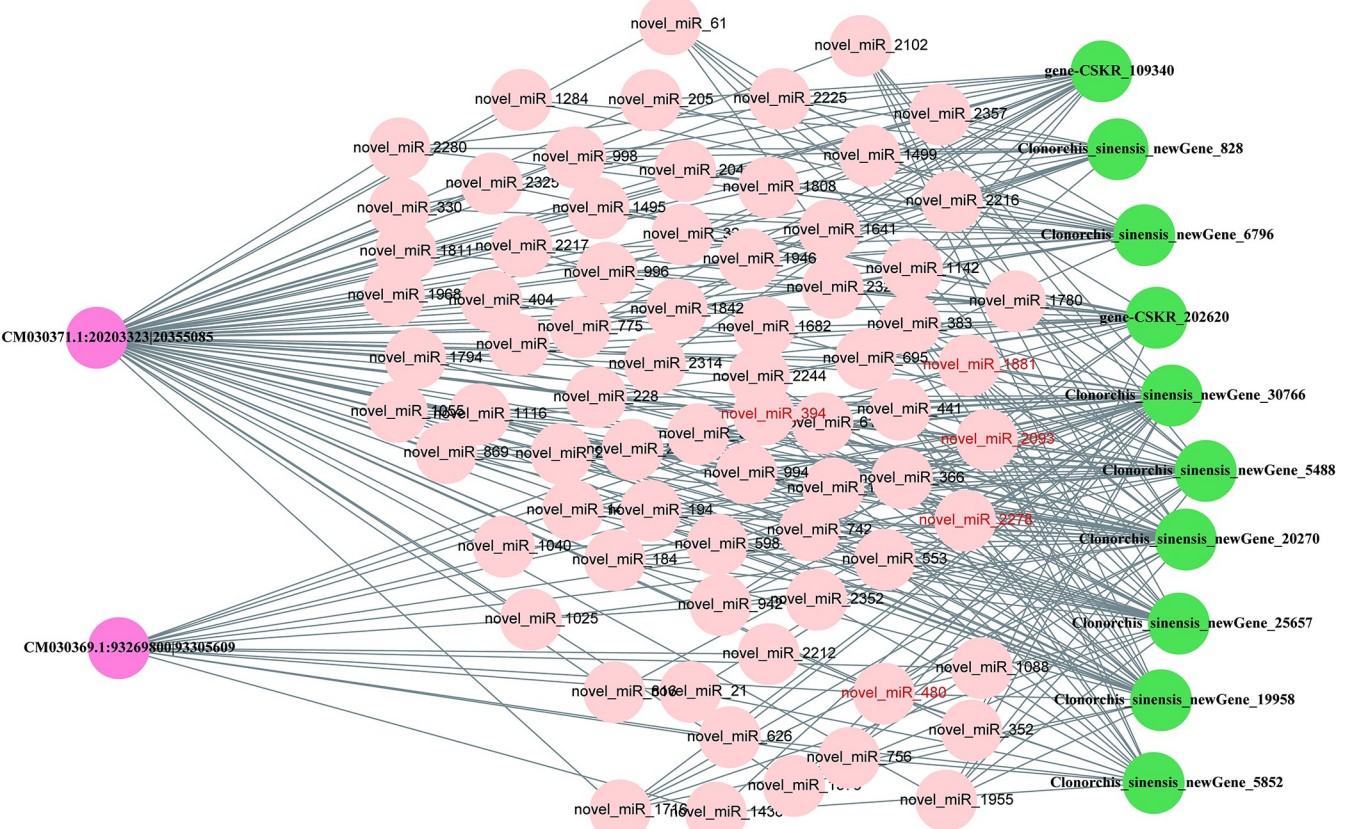

**Fig 7. ceRNA network representation of computationally predicted binding interactions between circRNA, miRNAs and mRNAs in *C. Sinensis*.** This network represents an excerpt of the circRNA-miRNA-mRNA network detailed in S11 Table, showing only the circRNAs that coexpressed microtubule-related mRNAs in *C. sinensis*. The top5 degree miRNAs are novel_miR_394, novel_miR_1881, novel_miR_2093, novel_miR_2278, and novel_miR_480 in circRNA ceRNA. The circRNA nodes are represented by pink circles, the miRNA nodes are represented by tangerine circles, and the mRNA nodes are represented by green circles. Connections between nodes are represented by gray edges.

## Discussion

Clonorchiasis, a zoonosis caused by infection with *C. sinensis*, can lead to severe liver and gall-bladder diseases if the infection is prolonged [4]. To effectively combat this parasite in the future, we sequenced four different developmental stages of *C. sinensis* by using the whole transcriptome. This study reported on the sequencing, annotation, and comprehensive analysis of the whole transcriptome of *C. sinensis*. The establishment of a network encompassing various RNA analyses for reciprocal regulation represents a pioneering effort, which possesses immense significance for elucidating the biological functions that are inherent in the whole transcriptome of *C. sinensis*.

Compared with traditional sequencing techniques, next-generation sequencing has many advantages and is frequently used for gene sequencing analysis of parasites [25–27]. In this study, we identified 25,459 mRNAs, 2,384 miRNAs, 27,564 lncRNAs, and 3,696 circRNAs in *C. sinensis*. Because there are limited miRNA data on *C. sinensis* in public databases and there are no relevant data on lncRNAs and circRNAs, we can only compare the miRNA and lncRNA data of *C. sinensis* with published datasets of *Schistosoma mansoni*, *S. japonicum* and *F. hepatica* [28–30]. Through the comparisons, it was found that the homology of the miRNA sequence of *C. sinensis* and *S. japonicum* was high. In addition, by comparing the lncRNA

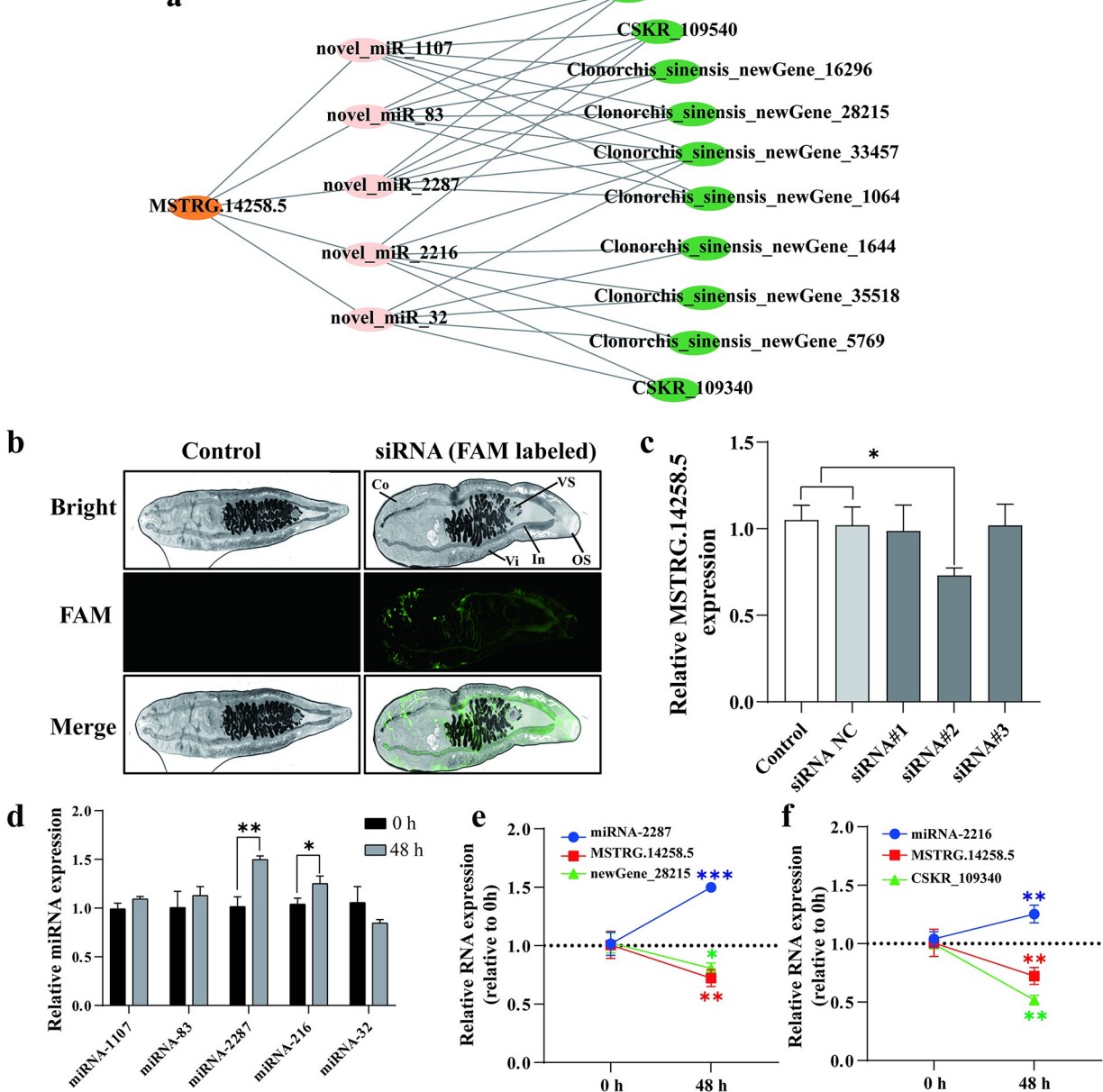

**Fig 8. Verification of the lncRNA (MSTRG.28240.1) ceRNA network in *C. sinensis*.** (**a**) Putative ceRNA regulatory network for lncRNA MSTRG.28240.1. (**b**) Electrotransformation of fam-labeled siRNA to *C. Sinensis*. Co is cortex, VS is ventral sucker, Vi is vitellaria, In is intestine and OS is oral sucker. (**c**) The relative expression of MSTR.28240.1 after electrotransformation of different siRNAs. (**d**) The relative expression of MSTRG.28240.1-targeted miRNA after 48 hours of electrotransformation of siRNA#2. (**e-f**) The relative expression of RNAs molecules with ceRNA regulatory network relationships in *C. sinensis* after 48 hours of interference with siRNA#2. Compared with indicated groups, * is for $P < 0.05$, ** is for $P < 0.01$, *** is for $P < 0.001$.

sequences of the BLASTn and RNAcentral databases, it was found that the lncRNA sequences of *C. sinensis* have low homology with lncRNAs of other flukes. These results indicate that miRNAs exhibit greater conservation between species than lncRNAs. Thus, if a secreted lncRNA chip of various parasites can be constructed in the future, the presence of parasitic infections can be specifically determined by detecting host fluids (such as serum and urine), such as miRNAs used for diagnostic biomarkers [31, 32]. Due to the lack of reports on

circRNAs from other flukes, the circRNA dataset is a useful repository that needs to be developed because many studies have shown that circRNAs play important roles in regulating body growth and disease occurrence and development.

Previous studies have highlighted distinctions between lncRNAs and mRNAs, and our results were consistent with these findings [28, 33, 34]. Specifically, lncRNAs exhibit shorter gene lengths, lower expression abundances, fewer exons, and a reduced number of isoforms per gene. This disparity has garnered considerable attention in molecular biology and bioinformatics, thus offering insights into the functional and evolutionary aspects of lncRNAs. LncRNAs are implicated in gene expression regulation through diverse mechanisms, including chromatin modification, transcription factor regulation, and miRNA induction [15, 35]. Due to their relatively modest expression levels, the cellular concentration of lncRNAs may not be sufficient for extensive protein synthesis. The evolutionary dynamics of lncRNAs could be influenced by distinct selection pressures, thus rendering them more susceptible to mutations than protein-encoding genes [36]. Consequently, this may manifest as shorter lengths, lower expression levels, and a scarcity of isoforms. LncRNAs actively engage in intricate cellular regulatory networks by interacting with other RNA molecules, including mRNAs and miRNAs [13]. Due to their short length and low expression, there are well suited for regulatory roles within these networks. These differences contribute to the complexity and adaptability of cellular mechanisms regulating gene expression. An in-depth understanding of these differences will improve our understanding of lncRNA functions in biological and disease processes and provide new research opportunities for the diagnosis and treatment of associated diseases.

The differential expression analysis utilized libraries with three biological replicates for the egg, metacercariae, excysted metacercariae and adult stages, representing different life stages of *C. sinensis*. In all stages, stage-specific mRNAs, lncRNAs, miRNAs, and circRNAs were identified. mRNAs exhibited the greatest number of DEGs, whereas circRNAs exhibited the lowest number of DEGs. Notably, the most significant disparities in gene expression were observed between the egg::metacercariae and excysted metacercariae::adult comparisons. This discrepancy can be attributed to the multifaceted developmental process of *C. sinensis* eggs progressing through miracidia, sporocysts, redia and cercariae to reach the metacercariae stage. Furthermore, different intermediate hosts during these stages contribute to significant variations in gene expression. Analysis of these DEGs revealed critical molecules influencing the growth and development of *C. sinensis*. In pairwise comparisons of DE-miRNAs, DE-lncRNAs, DE-circRNAs and DE-mRNAs at different stages of *C. sinensis*, the consistency in the proportion of DEGs indicates that the expression of ncRNAs (miRNAs, lncRNAs and circRNAs) mirrors the transcriptome dynamics of mRNAs. This further supports the hypothesis that noncoding RNAs play an important and stage-specific role in the parasite life cycle [28].

Through the GO-BP analysis, we found that the DEGs were mainly enriched in biological processes related to microtubule, with 42 genes associated with these processes. Notably, within the module, 47.6% (20/42) of the mRNA transcripts were related to the cytoskeleton, including tubulin alpha chain, dynein light chain, and hypothetical proteins. We randomly selected ten mRNAs for RT–qPCR validation, and the results indicated that the transcription levels of dynein light chain 2 and dynein light chain 4 gradually increased with the development of *C. sinensis*. Since dyneins are microtubule-based molecular motors that are responsible for various intracellular transport processes and the regulation of cellular structures [37, 38], thus, as the components of the dynein motor complex, DLC2/4 are essential to the transport of matter and energy. During the development of *C. sinensis*, demand for energy and matter is gradually increasing, we suggested that the expression levels of DLC2/4 should correspondingly increase. In another study, researchers found that expression of dynein light chain of *S. mansoni* is developmentally regulated and localized to the tegument in the

schistosomula, lung stage worms, and adult worms, but is not present in the cercariae or ciliated miracidia [39]. Thus, it is believed that DLC2/4 may play a certain role in the growth and development of *C. sinensis*. For *C. sinensis*, albendazole is an effective anthelmintic, and albendazole deworming is achieved by disrupting the microtubule structure within the parasite cells [40, 41]. Therefore, we treated *C. sinensis* with different concentrations of albendazole. The results indicated that with increasing albendazole concentration, the gene transcription levels of DLC2/4 gradually decrease. These indicate that DLC2/4 may be effective therapeutic targets for *C. sinensis* in the future.

Various studies have shown that noncoding RNAs, such as miRNAs, play crucial roles in organismal growth and development, disease onset and progression, and disease diagnosis and treatment [42, 43]. Moreover, miRNAs act as pivotal links between lncRNAs and circRNAs, thus connecting them with mRNAs through the ceRNA hypothesis [44, 45]. According to this hypothesis, lncRNAs and circRNAs (known as MREs) contain miRNA binding sites, thus enabling them to function as miRNA sponges. This "sponge-like" phenomenon is believed to foster competition between miRNAs and homologous mRNA targets, thereby enabling precise control of miRNA regulation of mRNA target transcripts. Thus, based on the ceRNA hypothesis, we selected lncRNAs and circRNAs that coexpressed with 42 microtubule-associated mRNAs to construct lncRNA-miRNA-mRNA and circRNA-miRNA-mRNA networks. As genes with similar expression trends generally share common biological functions, we ultimately identified three lncRNAs and two circRNAs for constructing the ceRNA network. In addition, the validation of the lncRNA ceRNA networks indicated that the ceRNA network was involved in the regulation of *C. sinensis*, thus suggesting that lncRNAs can be used as candidate drug targets by RNAi [46]. In the clinical practice, givosiran, an RNAi therapeutic drug by targeting aminolevulinic acid synthase 1 (ALAS1) has been developed for the treatment of acute hepatic porphyria (AHP) [47]. In the future, with more key lncRNAs are identified, we can interfere with the expression of lncRNAs through siRNA, thereby affecting the growth and development of *C. sinensis* and achieving the goal of reducing *C. sinensis*.

This study represents the first comprehensive exploration of the whole transcriptome of *C. sinensis*, which systematically elucidate the dynamic expression of miRNAs, lncRNAs, and circRNAs at different developmental stages of *C. sinensis*. Notably, different RNA molecules exhibit stage-specific regulatory functions, which providing insights into the intricate developmental processes of *C. sinensis*. Additionally, this study identified microtubule-related genes as potential regulators in *C. sinensis*, and the coexpression of lncRNAs and circRNAs with microtubule-associated genes suggested that they may be involved in the growth and developmental regulation of *C. sinensis* through a ceRNA regulatory mechanism, but further verification is needed in the future. This discovery expands the spectrum of research methodologies for noncoding RNAs in *C. sinensis*, effectively addresses gaps in lncRNA and circRNA exploration, and has implications for advancing diagnostic and therapeutic strategies for *C. sinensis* infections.

## Supporting information

**S1 Fig. Expression of miRNA and mRNA in different stages of *C. sinensis*.** RT−qPCR results for the relative expression of miRNA and mRNA in egg (EGG), metacercariae (M), excysted metacercariae (EM) and adult (AD) of *C. sinensis*.
(TIF)

**S2 Fig. Transcription levels of lncRNAs, miRNAs and mRNAs after MSTRG.14258.5 was interfered with siRNA#2.**
(TIF)

**S1 Table. The *Clonorchis sinensis* primers that were used for RT–qPCR.**
(DOCX)

**S2 Table. siRNA sequence of MSTRG.14258.5.**
(XLSX)

**S3 Table. Differential expression of mRNAs at four different stages of *Clonorchis sinensis*.**
(XLSX)

**S4 Table. Differential expression of lncRNAs at four different stages of *Clonorchis sinensis*.**
(XLSX)

**S5 Table. Differential expression of miRNAs at four different stages of *Clonorchis sinensis*.**
(XLSX)

**S6 Table. Differential expression of circRNAs at four different stages of *Clonorchis sinensis*.**
(XLSX)

**S7 Table. Forty-two microtubule-related mRNAs and coexpressed noncoding RNAs (lncRNAs and circRNAs) were identified in *Clonorchis sinensis*.**
(XLSX)

**S8 Table. LncRNA-miRNA-mRNA ceRNA networks in *Clonorchis sinensis*.**
(XLSX)

**S9 Table. LncRNA-miRNA-mRNA ceRNA networks (lncRNA coexpressed with microtubule-related genes) in *Clonorchis sinensis*.**
(XLSX)

**S10 Table. CircRNA-miRNA-mRNA ceRNA networks in *Clonorchis sinensis*.**
(XLSX)

**S11 Table. CircRNA-miRNA-mRNA ceRNA networks (circRNA coexpressed with microtubule-related genes) in *Clonorchis sinensis*.**
(XLSX)

## Acknowledgments

Drs. Xiaoxia Wu and Xiaoxiao Ma at Key Laboratory for Zoonosis Research of the Ministry of Education, Institute of Zoonosis, Jilin University, are thanked for providing some of the *Clonorchis sinensis* specimens used in this study.

## Author Contributions

**Conceptualization:** Mingyuan Liu, Xiaolei Liu, Chen Li, Bin Tang.

**Formal analysis:** Mingyuan Liu, Xiaolei Liu, Chen Li, Bin Tang.

**Funding acquisition:** Mingyuan Liu, Xiaolei Liu, Chen Li, Bin Tang.

**Methodology:** Yangyuan Qiu, Cunzhou Wang, Jing Wang, Qingbo L. V.

**Project administration:** Xiaolei Liu.

**Resources:** Lulu Sun, Yaming Yang.

**Writing – original draft:** Yangyuan Qiu, Cunzhou Wang.

**Writing – review & editing:** Xiaolei Liu, Chen Li, Bin Tang.

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
