## [Decision Letter · Decision Letter 0]

5 Mar 2024

Dear Dr. Tang,

Thank you very much for submitting your manuscript "Revealing the dynamic whole transcriptome landscape of Clonorchis sinensis : insights into the regulatory roles of non-coding RNAs and microtubule-related genes in development" for consideration at PLOS Neglected Tropical Diseases. As with all papers reviewed by the journal, your manuscript was reviewed by members of the editorial board and by several independent reviewers. In light of the reviews (below this email), we would like to invite the resubmission of a significantly-revised version that takes into account the reviewers' comments. 

We cannot make any decision about publication until we have seen the revised manuscript and your response to the reviewers' comments. Your revised manuscript is also likely to be sent to reviewers for further evaluation.

Sincerely,

Aysegul Taylan Ozkan, M.D., Ph.D.,

Academic Editor

Richard Bradbury

Section Editor

Reviewer's Responses to Questions

**Key Review Criteria Required for Acceptance?**

**Methods**

-Are the objectives of the study clearly articulated with a clear testable hypothesis stated?

-Is the study design appropriate to address the stated objectives?

-Is the population clearly described and appropriate for the hypothesis being tested?

-Is the sample size sufficient to ensure adequate power to address the hypothesis being tested?

-Were correct statistical analysis used to support conclusions?

-Are there concerns about ethical or regulatory requirements being met?

Reviewer #1: Minor revision.

The author isolated C. sinensis from Pseudorasbora parva and then used rabbits as experimental subjects. Rabbits were infected with 500 metacercariae per rabbit, and parasites were collected at the 7th, 14th, and 56th days post-infection. How were the samples retrieved? It is suggested here that the author provide supplementary information regarding the method used to isolate parasites from rabbits.

Reviewer #2: (No Response)

**Results**

-Does the analysis presented match the analysis plan?

-Are the results clearly and completely presented?

-Are the figures (Tables, Images) of sufficient quality for clarity?

Reviewer #1: Major revision.

The authors focus on bioinformatics analyses in the manuscript, while the presentation of experimental data is comparatively limited. While the use of high-throughput sequencing and bioinformatics tools is essential for gaining insights into transcriptomic profiles, it is equally important to support these findings with experimental validation. It would be beneficial to complement these findings with experimental validation. For example, the study could benefit from functional experiments such as gene knockout, knockdown, or overexpression studies to directly assess the roles of microtubule-related genes and non-coding RNAs in C. sinensis development. Including such experimental data would provide a more comprehensive understanding of the mechanisms involved.

In the results section of this article, the authors merely present data without providing summaries and conclusions for each result. It is advisable to include concise 1-2 summaries of the conclusions drawn from each result.

Reviewer #2: (No Response)

**Conclusions**

-Are the conclusions supported by the data presented?

-Are the limitations of analysis clearly described?

-Do the authors discuss how these data can be helpful to advance our understanding of the topic under study?

-Is public health relevance addressed?

Reviewer #1: Minor revision.

It is worth highlighting the practical implications of these findings, including their potential applications in the diagnosis and treatment of C. sinensis infections. Given the significant impact of C. sinensis on human hepatobiliary diseases, the insights gained from this study could potentially contribute to the development of more effective diagnostic methods and therapeutic interventions for this parasitic infection. Discussing these practical applications will not only underscore the relevance of the research but also its potential to make a meaningful difference in public health.

Reviewer #2: (No Response)

**Editorial and Data Presentation Modifications?**

Reviewer #1: Minor revision.

Reviewer #2: (No Response)

**Summary and General Comments**

Reviewer #1: The authors describes research on Clonorchis sinensis, a food-borne parasite causing hepatobiliary diseases, including cholangiocarcinoma. Transcriptome sequencing and the chromosomal-level genome analysis were carried out to study RNA regulation during C. sinensis development. They identified thousands of differentially expressed RNAs across 37 developmental stages and highlighted the critical role of microtubule-related biological processes in the growth and development of C. sinensis. Two genes, DLC2 and DLC4, showed increased expression during development, and regulatory networks involving lncRNA-miRNA-mRNA and circRNA-miRNA-mRNA interactions were constructed. Overall, I find the study to be promising and potentially valuable in advancing our understanding of C. sinensis development. However, I do have some concerns regarding the balance between theoretical analysis and experimental data in the manuscript.

Reviewer #2: 1.Fig 1. 

what are NIRI..... in panel d, Plz describe in legend. 

Line 565, “on the” should added between “circular RNA” and “reference....”

Panel f should be redraw because that orange column was beyond the left boundary. 

2.Fig 2. 

Line 572, in legend lncRNs and mRNs meaned lncRNAs and mRNAs?

Line 573 and Line 574, ”at” should be added before “different” 

Panel c, EGG, M, EM, and AD meaned what? It should be interpreted in the legend.

3.Fig 3, The comparision between what with what shouled be pointed out in Panel a, b,c and d. 

4.Fig 4, “Enrichment of Gene Ontology Biological Process (GO-BP) of differential expression mRNAs in four developmental stages of C. sinensis” will be more suitable for the subtitle of the legend.

5.Fig 5, The abbreviations of 33KD, DIC, DLC1, DLC2,....meaned what? It should be interpreted in the legend.

6.Fig 6 and 7, “the more critical miRNAs (TOP5 degree) were marked in red” should be included in the legends.

7.In all figure legends, “Clonorchis sinensis” could be replaced by “C. sinensis”. 

8.Fig S1 legend should be provided.

9.Why parasites were collected at the 7th, 14th, and 56th days post infection?

10.47.6% (20/42) of mRNA transcripts are related to the cytoskeleton, including 6 tubulin alpha/ beta chains, 1 dynein intermediate chain, and 5 hypothetical proteins. The author randomly selected 8 mRNAs for qPCR validation. Why tubulin alpha/ beta chains were not included?

11.Line 453, 459, and 465, “dynein intermediate chain 2 and dynein intermediate chain 4” should be “dynein light chain 2 and dynein light chain 4”. The discussion about the gradually increase of DLC 2/4 is a little unconvincing.

Line 45-48, “This study suggested that through whole transcriptome sequencing the context of microtubule regulation might play a role in the development and growth of C. sinensis.” will be better than now.

12.The MS would be greatly improved if one ceRNA regulatory mechanism could be verified. 

13.The lauguage should be checked by native speaker.

PLOS authors have the option to publish the peer review history of their article (what does this mean?). If published, this will include your full peer review and any attached files.

Reviewer #1: No

Reviewer #2: No
---

## [Decision Letter · Decision Letter 1]

18 May 2024

Dear Dr. Tang,

Thank you very much for submitting your manuscript "Revealing the dynamic whole transcriptome landscape of Clonorchis sinensis : insights into the regulatory roles of noncoding RNAs and microtubule-related genes in development" for consideration at PLOS Neglected Tropical Diseases. As with all papers reviewed by the journal, your manuscript was reviewed by members of the editorial board and by several independent reviewers. In light of the reviews (below this email), we would like to invite the resubmission of a significantly-revised version that takes into account the reviewers' comments. 

We cannot make any decision about publication until we have seen the revised manuscript and your response to the reviewers' comments. Your revised manuscript is also likely to be sent to reviewers for further evaluation.

Sincerely,

Aysegul Taylan Ozkan, M.D., Ph.D.,

Academic Editor

Eva Clark

Section Editor

Reviewer's Responses to Questions

**Key Review Criteria Required for Acceptance?**

**Methods**

-Are the objectives of the study clearly articulated with a clear testable hypothesis stated?

-Is the study design appropriate to address the stated objectives?

-Is the population clearly described and appropriate for the hypothesis being tested?

-Is the sample size sufficient to ensure adequate power to address the hypothesis being tested?

-Were correct statistical analysis used to support conclusions?

-Are there concerns about ethical or regulatory requirements being met?

Reviewer #1: The study design is appropriate and sufficient to address the stated objective.

Reviewer #3: (No Response)

**Results**

-Does the analysis presented match the analysis plan?

-Are the results clearly and completely presented?

-Are the figures (Tables, Images) of sufficient quality for clarity?

Reviewer #1: The results are clearly and completely presented, of sufficient for clarity.

Reviewer #3: (No Response)

**Conclusions**

-Are the conclusions supported by the data presented?

-Are the limitations of analysis clearly described?

-Do the authors discuss how these data can be helpful to advance our understanding of the topic under study?

-Is public health relevance addressed?

Reviewer #1: The conclusions are supported by the data presented and public health relevance is addressed.

Reviewer #3: (No Response)

**Editorial and Data Presentation Modifications?**

Reviewer #1: 1.Line 189, "sRNA" needs to be checked for accuracy.

2.Line 260, should it be for |r|?

3.Line266-267, should it be C. sinensis of 56 day?

4.Line396, what does 12 refer to? It needs to be clearly stated.

Reviewer #3: (No Response)

**Summary and General Comments**

Reviewer #1: This paper not only described the expressions and interactions between mRNAs and ncRNAs of C. sinensis on the different stages of development, but also verified the interactions betweeen lncRNA and miRNA. The sequencing data analysis and experiment designed were sufficient to support the object of the study, and the results were solid and comprehensive.

Reviewer #3: I respect the efforts of the authors for this study, and the authors try to reveal the RNA characteristics of C. sinensis at four developmental periods through comparative transcriptomics, and construct the ceRNA network based on these data. However, the article is flawed and needs to be addressed and revised for clarification.

Major comments:

1) mRNA codes the protein and is the central of ceRNA network. So why DE-mRNA was not chosen for quantitative validation, what’s the reason? 

2) In Results section (L359): ‘MSTRG.14258.5, MSTRG.19374.4, and MSTRG.9295.1 were identified as related to microtubule regulation lncRNAs’. These three lncRNAs were only predicted to be related to microtubule regulation, not identified in molecular experiment. Similar description in L497-499, ‘The co-expression of lncRNAs and circRNAs with microtubule-related genes suggest their participation in the growth and development regulation of C. sinensis through ceRNA regulatory mechanism’. This conclusion is not justified by the fact that the authors only constructed a ceRNA network based on sequencing results and software predictions, and did not experimentally verify that microtubule-related genes are involved in the growth and development of C. sinensis through the ceRNA regulatory mechanism. 

3)The graphs were not high enough resolution and they were too blurry.

Minors: 

L4-5. The number of authors does not match. the number of authors is 10 in L4-5, while L25-26, the number of authors is 11, there is an extra “Xiaoxiao Ma” among the co-authors, and the contribution of “Xiaoxiao Ma” is not stated in L524 “Authors' contributions”. 

L5. Wrong corresponding author name. L5. And L26, it was ‘Bin Tang‘, but in L20, it was written Bing Tang. 

L126. ‘the dynamic whole transcriptome landscape of Clonorchis sinensis’, however, the number of samples and biological replicates were not stated in Materials and Methods.

L172. Reference genome. The full name, and version of reference genome is required. 

L188. What’s the meaning of sRNA here, according to the context, it seems to be miRNA.

L201. The name of the package should be DESeq2, please check and verify. 

L210, The version of software Goseq, KOBAS (l218) is required. 

L229. What is the specific role of albendazole, what is the connection between the results of albendazole treatment and the results of differentially expressed genes at different developmental periods in this study.

L288. Fig.2E, L289. Fig.2e. 

L292. The number of sequencing groups should be explicitly stated in the methods; meanwhile, the L292 differential gene analysis, should state that these differential RNA molecules should be the total number of RNA molecules compared across groups.

L294. “|log2FC| > 1”, the correct one is in line204，“log2(Fold change) > 1”，2 is supposed to be subscripted in L294. 

L294. 1)The threshold P<0.001 used here is inconsistent with that in the Methods (L203, adjusted P value < 0.001); 2)there is a problem with the threshold line in the vertical coordinate of Figures 3a-d (especially b); 3)and it is not stated which two groups are being compared for the differential expression analysis of the different RNAs in Figures 3a-d. 4)If it is different developmental periods, then the volcano plots for differential expression analysis of each RNA should not be only one Figure, and the other results should be put in the supplementary file.

L307, 313, 315. According to the context, it is supposed to be Fig. 3e, not 2e. 

L546. Questioning the authenticity of its animal ethics approval letter. L546, Ethical approval and consent to participate, ‘All mice were handled in strict accordance with the People's Republic of China Animal Ethical Procedures and Guidelines’, while in L133, 224, 225 state that the experimental animals in this study were ‘rabbits’, not mice in the animal ethics approval letter.

L572. It was supposed to be mRNAs, not ‘mRNs’.

L574. ‘at’ was omitted before ‘different developmental stages’. 

L593. It was adult and excysted metacercariae in C. sinensis in (b) in L591, and here it was adult and excysted metacercariae in C. sinensis. Please check and revise the group names clearly.

Figures:

Fig2c, 2d are labeled “***” and “ns”, but their meaning is not explained in the figure notes of Fig2. 

Fig3. L577.The title of Fig.3 does not match the content of the fig3, which is labeled with the thresholds of differentially expressed mRNAs, lncRNAs, miRNAs, and circRNAs as “P < 0.001”, whereas the fig3 note states “P < 0.01”. In Fig3, b and d. The threshold lines didn’t match the plot. 

Fig5c. Figure 5 is labeled “***” and “*”, but the title does not indicate what they mean.

PLOS authors have the option to publish the peer review history of their article (what does this mean?). If published, this will include your full peer review and any attached files.

Reviewer #1: No

Reviewer #3: Yes: Xiaolong Kang
---

## [Decision Letter · Decision Letter 2]

23 Jun 2024

Dear Dr. Tang,

We are pleased to inform you that your manuscript 'Revealing the dynamic whole transcriptome landscape of Clonorchis sinensis : insights into the regulatory roles of noncoding RNAs and microtubule-related genes in development' has been provisionally accepted for publication in PLOS Neglected Tropical Diseases.

Best regards,

Aysegul Taylan Ozkan, M.D., Ph.D.,

Academic Editor

Eva Clark

Section Editor

Reviewer's Responses to Questions

**Key Review Criteria Required for Acceptance?**

**Methods**

-Are the objectives of the study clearly articulated with a clear testable hypothesis stated?

-Is the study design appropriate to address the stated objectives?

-Is the population clearly described and appropriate for the hypothesis being tested?

-Is the sample size sufficient to ensure adequate power to address the hypothesis being tested?

-Were correct statistical analysis used to support conclusions?

-Are there concerns about ethical or regulatory requirements being met?

Reviewer #3: (No Response)

**Results**

-Does the analysis presented match the analysis plan?

-Are the results clearly and completely presented?

-Are the figures (Tables, Images) of sufficient quality for clarity?

Reviewer #3: (No Response)

**Conclusions**

-Are the conclusions supported by the data presented?

-Are the limitations of analysis clearly described?

-Do the authors discuss how these data can be helpful to advance our understanding of the topic under study?

-Is public health relevance addressed?

Reviewer #3: (No Response)

**Editorial and Data Presentation Modifications?**

Reviewer #3: (No Response)

**Summary and General Comments**

Reviewer #3: The authors have addressed all comments clearly.

PLOS authors have the option to publish the peer review history of their article (what does this mean?). If published, this will include your full peer review and any attached files.

Reviewer #3: No

---

## [Editor Report · Acceptance letter]

7 Jul 2024

Dear Dr. Tang,

We are delighted to inform you that your manuscript, "Revealing the dynamic whole transcriptome landscape of Clonorchis sinensis : insights into the regulatory roles of noncoding RNAs and microtubule-related genes in development," has been formally accepted for publication in PLOS Neglected Tropical Diseases.

Best regards,

Shaden Kamhawi

co-Editor-in-Chief

Paul Brindley

co-Editor-in-Chief
